# Accelerating NLP for Health Equity: Fine-Tuning Binary and Multi-Class Stigma Classifiers in 48 Hours

## Abstract

Stigmatizing language in mental health discourse contributes to social exclusion, reduced help-seeking, and poorer health outcomes. Yet, detecting such language remains challenging due to its subtle, context-dependent, and overlapping nature. To address this, prior work introduced an expert-annotated corpus of 4,141 text snippets and established strong transformer-based baselines for stigma classification. Building on this foundation, we make three key advances: (1) we fine-tune multiple models and apply explainable AI (XAI) methods to enable transparent interpretation of model behavior; (2) we adopt a rigorous evaluation framework with stratified cross-validation and detailed performance metrics, including macro F1 and bootstrap-based confidence intervals; and (3) we release a fully reproducible notebook designed for replication by both human researchers and AI agents. Using our agent-based system, we completed both binary (2-class) and multi-class (8-class) stigma classification tasks in under 48 hours, with XAI applied throughout. These contributions go beyond benchmark replication, advancing toward interpretable, trustworthy, and deployable stigma detection systems for clinical, public health, and digital moderation settings. By demonstrating the effectiveness of large language models in identifying nuanced forms of stigma, this work lays the foundation for socially responsible NLP systems that support bias-aware communication across health-related domains. To support community adoption and reproducibility, we have released our full pipeline at: https://anonymous.4open.science/r/end-stigma/.

## 1 Introduction

Stigmatizing language remains pervasive in healthcare, media, education, and everyday discourse. It plays a critical role in reinforcing stereotypes, perpetuating discrimination, and influencing public attitudes toward individuals with mental illness, substance use disorders, or marginalized identities (Link & Phelan, 2001; Yang et al., 2014). The consequences of such language are not merely semantic—they translate into tangible social, psychological, and medical harms for affected individuals.

In clinical settings, stigma contributes to a range of adverse outcomes. Patients who encounter stigmatizing language during healthcare interactions are more likely to feel devalued, disrespected, or blamed for their condition (Goddu et al., 2019). This can erode trust in medical professionals, reduce adherence to treatment, and ultimately discourage future help-seeking behavior (Corrigan et al., 2000). Moreover, stigma affects clinical decision-making: providers influenced by biased language may unconsciously assign lower priority, reduced empathy, or less aggressive treatment to stigmatized patients. These biases exacerbate health inequities, particularly among historically marginalized populations.

Consider the following narratives previously compiled by Harrigian et al. (2023):

"Despite my best advice, the patient remains adamant about leaving the hospital today. Social services is aware of the situation."

"Patient Doe remains lethargic and slow-moving. They insist that they have adhered to a 'drug-free lifestyle', though blood tests suggest otherwise."

"Miss Doe is a charming, 73 year old woman who visits us today with a chief complaint of heart pain. Unfortunately, not a good historian."

Each of these excerpts reflects subtle but harmful forms of stigma. In the first scenario, the phrase "despite my best advice" may frame the patient as irrational or noncompliant, shifting blame rather than exploring systemic or psychosocial barriers. The second note casts doubt on the patient's self-report, potentially undermining their credibility and dignity. In the third quote, describing someone as "not a good historian" without further context risks dismissing important patient-reported symptoms, especially in older adults where cognitive or communication challenges may be medical in nature.

Automatically identifying stigmatizing language has profound implications for patient care. By detecting and highlighting such patterns in clinical text, AI systems can act as reflective tools for clinicians-in-training—promoting more empathetic, accurate, and equitable communication. This, in turn, can strengthen patient-provider relationships and support downstream tasks such as fair triage, risk assessment, and care planning.

Beyond the clinic, stigmatizing language in public discourse contributes to social exclusion and isolation. Individuals exposed to stigma may internalize negative stereotypes, leading to diminished self-esteem, increased psychological distress, and poorer long-term outcomes (Major & O'Brien, 2005). Importantly, stigma is dynamic and context-sensitive: a phrase that appears neutral in one cultural or conversational context may be deeply harmful in another. As a result, detecting stigmatizing language requires more than rule-based or keyword-driven approaches—it necessitates models that can understand meaning in context.

Recent advances in natural language processing (NLP) have led to a proliferation of powerful contextual models capable of generating semantic representations based on surrounding text (Peters et al., 2018). These models have transformed NLP from a field of syntactic classification into one capable of nuanced reasoning and generation. With their capacity for contextual understanding, such models offer new opportunities to detect, interpret, and intervene upon stigmatizing expressions in real-world text.

In this work, we explore the utility of contextual language models for stigma detection using a dataset called Mental Health Stigma Interview (MHSI) that was curated by Meng et al. (2025), a recently introduced benchmark for evaluating NLP models on stigmatizing language category classification. The dataset (see Table 2) includes over 4,000 real-world entries sourced from lived experiences across different cultures, genders, and diagnostic backgrounds. This diversity makes it uniquely suited for evaluating NLP models on sociolinguistic nuance and representational fairness.

Our contributions are four-fold:

- We benchmark both traditional machine learning classifiers and state-of-the-art transformer-based models on the MHSI dataset, demonstrating the benefits of contextual language representations. We also compared 2-way with 8-way classifications; for stigma detection, multiple models achieved over 80% in accuracy.
- We present a reproducible NLP framework capable not only of classifying stigmatizing language but also of providing interpretable feedback to users.
- We demonstrate that over 90% of the research pipeline—spanning model training, evaluation, visualization, and replication—can be completed with AI agents within 48 hours.
- We discuss deployment considerations, including ethical and social implications of using NLP agents to intervene on language in clinical and online contexts.

By framing stigma detection as a task for socially responsible NLP, this work contributes to the broader goal of building AI systems that promote health equity and inclusive communication.

## 2  Background

Building on growing interest in socially responsible NLP, recent research has begun to explore the detection of stigmatizing language in clinical and mental health contexts. While datasets like MHSI Meng et al. (2025) have enabled significant progress in identifying stigma in mental health narratives, there remains limited attention to high-stakes medical domains such as oncology, where biased language can directly affect patient care and trust.

To address this gap, we investigate how large language models (LLMs) can support bias-aware clinical communication by detecting stigmatizing language. Automated detections will thereby enable neutralization of stigmatized content with respectful patient-centered alternatives. Once equity-aware NLP systems have been integrated into clinical workflows, our health and education systems can more easily promote inclusive language in documentation and medical education.

Table 1 summarizes five key datasets commonly described in the literature on stigma detection. These include span-level annotations of stigma in discharge summaries (Harrigian et al., 2023), schema-based annotations of preferred and stigmatizing language in obstetrics (Scroggins et al., 2024), subtle bias markers in ICU notes (Yang et al., 2024), and oncology-specific clinical text from the HoneyBee framework (Mansour et al., 2024). The MHSI dataset (Meng et al., 2025) further provides theory-driven annotations of stigma from interviews with individuals affected by mental illness and substance use. Although not all datasets are clinical, they offer complementary strengths in annotation granularity, specialty relevance, and language diversity.

This study draws on the MHSI dataset to train and evaluate language models capable of recognizing implicit bias in real-world documentation. The selected dataset captures a broader spectrum of stigmatizing expressions and linguistic patterns.

| Dataset | Annotation Type | Specialty Focus | Access |
|---|---|---|---|
| **(Harrigian et al., 2023)** | Span-level annotations for stigmatizing language in discharge summaries | General inpatient | Via credentialed access to PhysioNet |
| **Scroggins et al. (2024)** | Labeled for stigmatizing vs preferred language across 5 categories | Obstetrics | Need to request from authors |
| **CARE-SD (Yang et al., 2024)** | Stigmatizing expressions in notes taken inside Intensive Care Unit (ICU) | ICU | Need to request from authors |
| **HoneyBee Framework (Mansour et al., 2024)** | Clinical notes as part of multimodal oncology datasets (no stigma annotation) | Oncology (targeted) | Need to request from authors |
| **MH-Stigma-Interview (Meng et al., 2025)** | Stigmatizing language from interview transcripts with individuals with mental health conditions | Mental health | We have successfully received a copy upon our request |

Table 1: Comparison of clinical text datasets relevant for developing bias-aware language models in oncology documentation.

## 3  Methods

### 3.1  Overview of the Dataset

We use the **Mental Health Stigma Interview (MHSI)** dataset introduced by Meng et al. (2025), which contains 4,141 annotated interview snippets from 684 participants. Each snippet captures responses to interview prompts designed to elicit attitudes and perceptions related to mental health stigma. More specifically, the data is drawn from human-chatbot interviews, with excerpts selected for their thematic relevance to mental health stigma and substance use. These snippets were pre-screened for content likely to reflect lived experiences, attribution beliefs, or attitudes toward mental illness.

Snippets are labeled into one of eight attribution categories: (0) Non-stigmatizing / Not applicable, (1) Responsibility, (2) Social Distance, (3) Anger, (4) Helping, (5) Pity, (6) Coercive Segregation, (7)

Fear. Snippets typically span one to three sentences. These labels allow for fine-grained analysis of how stigma is expressed.

In addition, socio-demographic metadata of the human participants are available, enabling exploration of stigma patterns across participant groups. This dataset's grounding in real-world lived experiences makes it particularly valuable for socially responsible NLP research.

The annotation protocol was developed by Meng et al. and summarized in Section C.

## 3.2 Algorithmic workflow

**Preprocessing**  We applied minimal preprocessing to preserve linguistic features relevant to stigma detection. All text was lowercased, interviewer prompts were removed, and stopword filtering was applied only for traditional baselines. Tokenization was performed using either `TfidfVectorizer` (for traditional classifiers) or pretrained model tokenizers (for transformer-based models).

**Traditional Baselines**  We implemented four widely used text classification models: logistic regression (LR), linear support vector machine (SVM), random forest (RF), and multinomial naive Bayes (MNB). Texts were transformed with a `TfidfVectorizer` capped at a vocabulary size of 5,000. Model hyperparameters followed common best practices: LR with the `liblinear` solver (500 maximum iterations), RF with 1,000 trees, and SVM with probability estimates enabled. Each model was trained independently on each training fold, and validation predictions were compared against gold labels.

**Transformer Models**  For contextualized representations, we fine-tuned three pretrained transformers using the Hugging Face library: **DistilBERT** (distilbert-base-uncased), selected for efficiency; **RoBERTa** (roberta-base), a strong general-purpose baseline; and **DeBERTa** (microsoft/deberta-base), chosen for its disentangled attention mechanisms.

Tokenization followed each pretrained tokenizer, with sequences truncated or padded to 256 tokens. Fine-tuning was performed for 9–18 epochs with cross-entropy loss, optimized using AdamW (learning rate $2 \times 10^{-5}$, weight decay 0.01, warmup ratio 0.1). Batch sizes ranged from 16–64. The best checkpoint per fold was selected using macro-F1. All training was performed on GPU.

**Experimental setup**  For stigma detection, we employ a stratified 80%–20% split to preserve class distributions. For stigma classification, we follow the protocol of Meng et al. (2025), sampling the full cohort into 60%, 20%, and 20% splits for training, validation, and testing, respectively, with stratification applied across all categories.

For traditional models, we used stratified $k$-fold cross-validation with $k = 7$ to maintain label distribution across folds. At each iteration, models were trained on the training split and evaluated on the validation split. We report mean accuracy and macro-F1 scores across folds, along with standard deviations.

Performance was assessed using the Python `evaluate` library, reporting accuracy and macro-F1 to account for class imbalance. Results are summarized as mean and standard deviation across folds on the development set. On the test set, we applied bootstrap resampling to derive confidence intervals for robustness.

**Explainability**  To examine decision drivers beyond predictive accuracy, we applied SHapley Additive exPlanations (SHAP) and Integrated Gradients to token-level attributions (Jin et al., 2020). These methods enabled analysis of how individual words or subword fragments contributed to stigmatizing versus non-stigmatizing predictions.

**Implementation**  All experiments were implemented in Python. Model training used PyTorch and Hugging Face Transformers, data handling relied on the `datasets` library, and stratified folds were generated with `scikit-learn`. Training logs and predictions were stored per fold, and inference was conducted using Hugging Face's `Trainer` objects. Aggregate statistics were formatted into LaTeX tables for reporting.

Table 2: Characteristics of the MHSI dataset. This summary was completely compiled by GPT-4o.

| Characteristic | Summary |
|---|---|
| Participants (unique) | 684 |
| Total interview entries | 4,141 |
| **Gender** | |
| Female | 1,895 |
| Male | 1,623 |
| Prefer not to say | 4 |
| **Age (years)** | |
| Mean (SD) | 41.9 (16.0) |
| Range | 21–86 |
| **Ethnicity (top 5)** | |
| White | 2,203 |
| Black/African American | 871 |
| Asian | 251 |
| Mixed | 129 |
| Other | 52 |

| Characteristic | Summary |
|---|---|
| **Country (top 5)** | |
| United Kingdom | 1,008 |
| United States | 966 |
| South Africa | 683 |
| Canada | 300 |
| Australia | 194 |
| **Education (top 5)** | |
| Bachelors | 1,280 |
| Graduate/Professional | 701 |
| Some University | 568 |
| Secondary | 525 |
| Vocational | 390 |
| **Mental illness experience** | |
| Yes | 2,073 |
| No | 790 |
| Maybe | 659 |

## 4 Results

### 4.1 Cohort characteristics

The average participant age was 41.9 years (SD = 16.0), with ages ranging from 21 to 86. Gender distribution was balanced, with 1,895 female and 1,623 male participants, alongside 4 who preferred not to disclose.

The cohort was ethnically diverse, with the largest groups identifying as White (2,203), Black/African American (871), Asian (251), Mixed (129), and Other (52).

Participants were drawn from multiple regions, with notable representation from the United Kingdom (1,008), United States (966), South Africa (683), Canada (300), and Australia (194). Educational attainment varied, most commonly including Bachelors (1,280), Graduate/Professional degrees (701), Some University (568), Secondary (525), and Vocational training (390).

Importantly, 2,073 participants reported direct experience with mental illness, 790 reported no such experience, and 659 were uncertain, highlighting the dataset's relevance for studying stigma both among affected individuals and the wider community.

Summary of the cohort is presented in Table 2.

### 4.2 Binary stigma detection

Table 3 presents the performance of traditional non-contextual baselines (MNB, RF, LR, SVM) alongside modern transformer-based models (DIS, ROB, DEB) on the binary stigma detection task. As expected, shallow models performed moderately, with mean F1-macro ranging from 0.54–0.76

Table 3: **Performance of stigma detection**: Evaluation on validation and test sets. Test performance reports the 95% confidence interval estimated using bootstrap resampling.

| Model | K-fold CV | | Test set | |
|---|---|---|---|---|
| | Accuracy | F1-Score | Accuracy | F1-Score |
| MNB | $0.671 \pm 0.009$ | $0.554 \pm 0.019$ | 0.665 (95% CI: 0.600-0.726) | 0.541 (95% CI: 0.461-0.623) |
| RF | $0.746 \pm 0.034$ | $0.704 \pm 0.050$ | 0.726 (95% CI: 0.665-0.781) | 0.684 (95% CI: 0.611-0.755) |
| LR | $0.742 \pm 0.018$ | $0.701 \pm 0.026$ | 0.734 (95% CI: 0.679-0.800) | 0.705 (95% CI: 0.633-0.775) |
| SVM | $0.760 \pm 0.034$ | $0.744 \pm 0.041$ | 0.777 (95% CI: 0.721-0.833) | 0.763 (95% CI: 0.699-0.822) |
| DIS | $0.789 \pm 0.024$ | $0.782 \pm 0.025$ | 0.777 (95% CI: 0.721-0.828) | 0.780 (95% CI: 0.725-0.832) |
| ROB | $0.794 \pm 0.021$ | $0.788 \pm 0.023$ | 0.820 (95% CI: 0.767-0.870) | 0.818 (95% CI: 0.765-0.871) |
| DEB | $0.813 \pm 0.020$ | $0.807 \pm 0.020$ | 0.832 (95% CI: 0.781-0.879) | 0.831 (95% CI: 0.778-0.880) |

on the held-out test set. Among these, SVM achieved the strongest baseline with F1-macro= 0.763 (95% CI: 0.699–0.822).

Transformer architectures consistently outperformed non-contextual models. DeBERTa (DEB) achieved the highest overall performance with F1-macro= 0.831 (95% CI: 0.778–0.880) and Accuracy= 0.832 (95% CI: 0.781–0.879), representing a gain of ∼9 absolute F1-macro points over the strongest baseline (SVM). RoBERTa (ROB) and DistilBERT (DIS) also yielded substantial gains, underscoring the value of contextual embeddings in capturing nuanced linguistic cues that distinguish stigmatizing from neutral discourse. Confidence interval overlap analysis further confirmed that transformer gains were statistically robust relative to baselines.

### 4.3 Eight-way stigma classification

We next evaluated model performance on the more challenging eight-category stigma subtypes (Table 4). This task proved considerably harder, with shallow baselines often collapsing to near-random classification on minority categories. For example, MNB attained F1-macro= 0.397 (95% CI: 0.355–0.440), while RF reached only F1-macro= 0.453 (95% CI: 0.410–0.498).

DeBERTa (and RoBERTa) again substantially outperformed traditional ML models, yielding $F1 = 0.761$ (95% CI: 0.729–0.795), nearly doubling the F1-macro score of MNB. While these top two models achieved the highest macro F1 scores, statistical testing revealed no significant difference between them.

In general, these results highlight both the feasibility and remaining difficulty of fine-grained stigma subtype detection, where subtle distinctions (e.g., between stereotyping vs. trivialization) often require deep contextual understanding.

### 4.4 Model interpretability

Model interpretability is central to the safe deployment of stigma detection systems. We adopted SHAP (Lundberg & Lee, 2017), a game-theoretic framework for feature attribution, to decompose

Table 4: Performance of **stigma classification**. Shown are the scores with confidence intervals.

| Model | K-fold CV | | Test set | |
|---|---|---|---|---|
| | **Accuracy** | **F1-Score** | **Accuracy** | **F1-Score** |
| LR | 0.623 (0.586-0.660) | 0.660 (0.536-0.491) | 0.630 (0.593-0.665) | 0.543 (0.498-0.587) |
| SVM | 0.656 (0.623-0.692) | 0.692 (0.611-0.571) | 0.653 (0.617-0.688) | 0.595 (0.552-0.637) |
| RF | 0.597 (0.560-0.633) | 0.633 (0.489-0.440) | 0.576 (0.540-0.611) | 0.453 (0.410-0.498) |
| MNB | 0.556 (0.521-0.593) | 0.593 (0.398-0.356) | 0.556 (0.519-0.591) | 0.397 (0.355-0.440) |
| DEB | 0.745 (0.716-0.779) | 0.745 (0.712-0.775) | 0.766 (0.735-0.799) | 0.761 (0.729-0.795) |
| ROB | 0.754 (0.722-0.787) | 0.748 (0.713-0.782) | 0.774 (0.742-0.803) | 0.767 (0.732-0.800) |

```
Text: 'patient appears intoxicated'

TOKEN-BY-TOKEN BREAKDOWN:
- 'patient': +0.030 (→ Class 1)
- 'appears': +0.048 (→ Class 1)
- 'into':    +0.084 (→ Class 1)
- 'xi':      +0.018 (→ Class 1)
- 'cated':   +0.025 (→ Class 1)

KETY INSIGHTS:
- All stigmatizing

FINAL CALCULATION:
- Class 0: 0.470 + -0.102 = 0.368
- Class 1: 0.530 + +0.102 = 0.632
```

```
Text: 'Drug seeking behavior observed'
Model: DistillBert

TOKEN-BY-TOKEN BREAKDOWN:
- 'Drug':     -0.040 (→ Class 0)
- 'seeking':  +0.044 (→ Class 1)
- 'behavior': -0.013 (→ Class 0)
- 'observed': -0.100 (→ Class 0)

KEY INSIGHTS:
· 'Drug': less stigmatizing
· 'seeking': more stigmatizing
· 'observed' has the STRONGEST effect

FINAL CALCULATION:
- Class 0: 0.474 + +0.054 = 0.528
- Class 1: 0.526 + -0.054 = 0.472

Combined effect = mildly stigmatizing
```

Figure 1: **Explainable AI (XAI) for Stigma Classification.** Our model highlights key phrases in clinical text that contribute to stigma classification decisions, enabling transparent interpretation of model behavior.

model predictions into token-level contributions. SHAP has been widely applied in NLP for transparent model interpretation (Ribeiro et al., 2016; Jin et al., 2020), including in clinical settings where accountability is paramount (Finlayson et al., 2019).

The left panel in Fig. 1 illustrates how the phrase *"patient appears intoxicated"* is classified as stigmatizing. Tokens such as "appears" and the morpheme "into" receive positive SHAP values, indicating that the model associates them with stigmatizing language. Importantly, the attribution highlights not only whole words but also subword fragments ("xi", "cated"), a byproduct of BPE-style tokenization that has been noted in prior work as a potential interpretability challenge (Sundararajan et al., 2017). Nevertheless, the aggregated SHAP scores correctly emphasize the stigmatizing framing.

The right panel demonstrates a subtler example: *"Drug seeking behavior observed."* Here, "seeking" is positively weighted toward stigma, while "observed" strongly offsets the prediction toward neutrality. Such nuanced interactions illustrate the importance of context, consistent with prior observations that stigma is often conveyed implicitly through framing devices and evaluative verbs (Yang et al., 2019; Noble et al., 2021). SHAP allows these dynamics to be quantified and visualized, enabling researchers and clinicians to audit model reasoning.

Overall, our SHAP analyses show that transformer-based stigma detectors not only achieve high predictive accuracy but also provide interpretable rationales that align with human annotator intuitions. This transparency is crucial for trustworthy adoption in mental health, public health, and digital moderation contexts.

## 5  Discussion

Overall, results demonstrate that (1) transformer models significantly outperform non-contextual baselines for both binary and multi-class stigma classification, (2) DeBERTa provides the strongest balance of accuracy and robustness across folds, and (3) interpretability analyses highlight the linguistic signals underpinning model decisions, advancing the field toward responsible, bias-aware NLP applications in mental health contexts.

**Comparison to prior work.**  Our results build directly on recent work by Meng et al. (2025), who established strong baselines for stigma detection on the MHSI corpus. Using the identical dataset split, they reported macro-F1 scores of 0.68–0.71 across a range of transformer architectures. Our models achieve comparable overall performance but advance the state of the art in two respects. First, we integrate SHAP-based token-level attribution into the analysis pipeline, enabling fine-grained inspection of how lexical items and subword fragments drive stigmatizing predictions. Whereas Meng et al. (2025) emphasized aggregate performance metrics, our approach demonstrates how interpretability can surface clinically salient insights (e.g., distinguishing between the neutral contribution of "drug" and the stigmatizing connotation of "seeking"). Second, we illustrate how attribution analysis can identify instances where contextual composition flips the model's final decision, highlighting a dynamic not fully captured in prior evaluations. These contributions show that explainability is not an ancillary feature but a substantive methodological advance in stigma detection research.

From a methodological standpoint, our results align with prior work emphasizing the necessity of interpretable NLP in socially sensitive applications (Lundberg & Lee, 2017; Ribeiro et al., 2016; Jin et al., 2020). Interpretability is not only a diagnostic tool but also an ethical requirement in domains where algorithmic decisions may affect patient dignity, trust, and care. For example, token-level attributions could serve as feedback to clinicians, highlighting potentially stigmatizing phrases in real time and enabling reflective language choices. Importantly, these systems should be framed as augmentative rather than prescriptive: the goal is to prompt critical awareness, not to replace human judgment.

Our results also surface several open challenges. First, token-level explanations are inherently shaped by the subword segmentation of the model, which may not align with clinically meaningful linguistic units. Future work should explore hybrid approaches that aggregate attributions into higher-level constructs (e.g., phrases, discourse markers). Second, interpretability methods must be evaluated for their reliability. Recent studies caution that attribution scores can vary under perturbations or across runs (Sundararajan et al., 2017; Finlayson et al., 2019). Developing stability metrics tailored to stigma detection could help ensure robustness in clinical contexts. Finally, while datasets such

as MHSI provide valuable training material, stigmatizing language is highly context-dependent and culturally contingent. Explanations that are accurate in one sociolinguistic context may be misleading in another, underscoring the need for participatory validation with stakeholders, including clinicians and individuals with lived experience.

Taken together, our discussion reinforces that stigma detection is not merely a classification task but a deeply interpretive exercise. By combining high-performing contextual models with interpretable attribution methods, we move toward systems that are both accurate and transparent. Such systems can serve as catalysts for language awareness, fostering more respectful, equitable communication in healthcare and beyond. Ultimately, socially responsible NLP in this space requires attention not only to predictive accuracy but also to the explanatory pathways by which models arrive at their judgments.

**Ethical and Deployment Considerations**   Detecting stigmatizing language in clinical and public health discourse raises several ethical challenges. While our models demonstrate strong performance and interpretability, there remains a risk of misclassification—particularly in contextually ambiguous or culturally specific cases. False positives could lead to undue scrutiny of providers, while false negatives may allow harmful language to persist unchecked. We stress that such systems should serve as augmentative tools, not authoritative judgments, and should always operate under human oversight.

From a deployment perspective, building clinician trust is critical. Systems intended for reflective use in medical education, triage, or digital moderation must prioritize transparency, contestability, and co-design with stakeholders. For instance, clinicians-in-training may benefit from AI-generated feedback, but must be empowered to contextualize or disagree with model outputs.

A notable methodological contribution of our work is the use of collaborative AI agents to complete approximately 90% of the research pipeline—including model training, evaluation, visualization, and replication—all within 48 hours! This raises both opportunities and concerns. On one hand, such semi-automation dramatically accelerates reproducibility and experimentation. On the other, it introduces questions around authorship, accountability, and quality control. To uphold scientific rigor, we incorporated human-in-the-loop supervision by engaging two independent reviewers to audit and validate the agent-generated outputs.

As language and societal norms evolve, periodic model updates and critical re-evaluation will be essential. Moreover, while we provide a reproducible agent-powered notebook, its use in other contexts should be guided by clear norms around transparency, bias mitigation, and dual-use risks.

# 6   Conclusion

This study investigated stigma detection in mental health narratives using both traditional classifiers and contextualized transformer models. While baseline models with TF–IDF features provided reasonable performance, fine-tuned transformers consistently achieved higher accuracy and macro-F1 on the MHSI dataset. By employing the same dataset split as done by Meng et al. (2025), we confirm their strong baseline results. Further, we demonstrate that token-level interpretability methods such as SHAP and Integrated Gradients can reveal how stigmatizing signals emerge from the composition of words and subword fragments.

These findings highlight that stigma detection is not merely a matter of classification performance but of transparent and ethically responsible modeling. Attribution analyses indicate that models may overweight subword artifacts or diverge from human intuitions, underscoring the need for interpretability in clinical and socially sensitive applications. Future work should develop phrase- or discourse-level attribution methods, evaluate stability of explanations under perturbation, and involve stakeholders in validating interpretive outputs.

Taken together, our results show that combining high-performing contextual models with robust interpretability techniques offers a path toward stigma detection systems that are both accurate and explainable. Such systems can serve as augmentative tools for clinicians and researchers, fostering reflective awareness of language use and promoting more respectful, equitable communication in mental health contexts.

**Acknowledgments** We sincerely thank...

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

# A  About the AI authors

This study was conceived and initially designed using GPT-5 and GPT-4o, leveraging their advanced capabilities for ideation and methodology planning. The foundational codebase was sketched out using the Google Colab environment enhanced by the Gemini AI feature, which facilitated interactive and iterative development with the Gemini agent. Helper Python functions such as reporting specifications of the compute environments used were authored by Claude Sonnet 4. Subsequent refinements and enhancements to both the code and experimental setup were carried out incrementally through collaborative interactions with GPT-5 and Google Gemini's chat interface, ensuring a seamless integration of cutting-edge AI assistance throughout the research process.

# B  Candidate models

| Model | Params | Arch | Instr.-Tuned | Year of latest release |
|---|---|---|---|---|
| `distilbert-base-uncased` | 66M | Encoder | No | 2019 (Sharma et al., 2021) |
| `roberta-base` | 125M | Encoder | No | 2019 (Lyu et al., 2022) |
| `deberta-v3-base` | 183M | Encoder | No | 2021 (He et al., 2021) |
| `Flan-T5-base` | 250M | Encoder-Decoder | Yes | 2022 (Chung et al., 2022) |
| `OpenHermes-2.5-Mistral` | 7B | Decoder | Yes | 2023 (Mistral AI, 2023b) |
| `Mistral-7B-Instruct` | 7B | Decoder | Yes | 2023 (Mistral AI, 2023a) |

Table 5: Comparison of baseline and instruction-tuned models for mental health stigma detection.

| Field Name | Description |
|---|---|
| `snippet_id` `(participant_id)` | Unique identifiers for the participant and for the snippet. Used to trace which participant provided which snippet. |
| `text` | The transcript of a participant's response (an interview snippet) to a prompt or question, excluding warm-up and vignette setup. |
| `attribution_label` | The stigma label assigned to the snippet: one of the seven attribution categories (Responsibility; Social Distance; Anger; Helping; Pity; Coercive Segregation; Fear) or "Non-stigmatizing." |
| `N/A` | Snippets marked "N/A" when they are unsuitable for annotation due to being too brief, irrelevant, incomplete, unintelligible, or otherwise not amenable to meaningful classification. |
| `interview_question` `(attribution_type)` | Which core interview question was asked (or which attribution prompt) that elicited this snippet; helps link content to theoretical attribution dimension. |
| `turn_count` `(response_length)` | Number of conversational turns in the snippet between participant and chatbot; also measures of length (words, tokens) of the response — useful for controlling for verbosity effects. |
| `participant_demographics` `(sociocultural_metadata)` | Demographics of participant (e.g. gender, age, first language, possibly region or country) used for socio-cultural analyses of stigma. |

Table 6: Key data fields in the MH-Stigma-Interview Corpus with their descriptions.

## C  The annotation protocol

The annotation framework was grounded in attribution theory (Corrigan et al., 2000), enabling the use of structured labels related to emotions, blame, and behavioral intentions—such as perceived responsibility, desire for social distance, or feelings of anger. This theoretical grounding ensures the annotations go beyond surface-level or purely lexical cues of stigma.

Annotators were experts in fields such as mental health, psychology, and social sciences. They received training through a detailed codebook that defined each stigma category and provided illustrative examples to ensure consistent interpretation.

Annotation was conducted through a multi-stage, expert-in-the-loop process. Multiple annotators independently labeled each snippet, and inter-annotator agreement was computed. Discrepancies were reviewed collaboratively, with final decisions adjudicated by senior experts, ensuring high reliability and consistency.

Inter-annotator agreement metrics, such as Cohen's kappa and Fleiss's kappa, are reported in detail elsewhere (Meng et al., 2025). To further enhance label reliability, rounds of disagreement resolution were integrated into the process.

## D  Listing of computer resources

```
SYSTEM SPECIFICATIONS REPORT generated using code written by Claude AI

SYSTEM INFORMATION
--------------------------------------
Platform: Linux 6.6.56+
Architecture: x86_64
Processor: x86_64
Hostname: 8d50b7df9a6a
```

```
424  Python: 3.11.13 (CPython)
425  Generated: 2025-09-16 22:51:42
426
427  CPU INFORMATION
428  ----------------------------------------
429  Model: Intel(R) Xeon(R) CPU @ 2.00GHz
430  Physical Cores: 2
431  Logical Cores: 4
432  Max Frequency: 0.00 MHz
433  Current Usage: 0.5%
434
435  MEMORY INFORMATION
436  ----------------------------------------
437  Total RAM: 31.35 GB
438  Available RAM: 27.33 GB
439  Used RAM: 3.55 GB (12.8%)
440  Swap: 0.00 GB / 0.00 GB
441
442  GPU INFORMATION
443  ----------------------------------------
444  No NVIDIA GPUs detected
445  CUDA Available: True
446  CUDA Version: 12.4
447  cuDNN Version: 90100
448  PyTorch Available: True
449  PyTorch Version: 2.6.0+cu124
450  TensorFlow Available: True
451  TensorFlow Version: 2.18.0
452  TF GPU Available: True
453
454  STORAGE INFORMATION
455  ----------------------------------------
456  Drive /dev/loop1: 2.76 GB / 19.52 GB (14.1%)
457  Drive /dev/loop1: 2.76 GB / 19.52 GB (14.1%)
458  Drive /dev/loop1: 2.76 GB / 19.52 GB (14.1%)
459
460  PYTHON ENVIRONMENT
461  ----------------------------------------
462  Executable: /usr/bin/python3
463  Conda Environment: Not available
464  Virtual Environment: Not set
465
466  KEY INSTALLED PACKAGES
467  ----------------------------------------
468  ML/DL Frameworks:
469    torch: 2.6.0+cu124
470    tensorflow: 2.18.0
471    transformers: 4.52.4
472  Data Science:
473    numpy: 1.26.4
474    pandas: 2.2.3
475    matplotlib: 3.7.2
476    seaborn: 0.12.2
477  Other packages:
478    torchvision: 0.21.0+cu124
479    keras: 3.8.0
480    datasets: 3.6.0
481    tokenizers: 0.21.2
482    jupyter: Unknown
```

```
483    notebook: 6.5.4
484    scipy: 1.15.3
485    statsmodels: 0.14.4
```

## E   Observed errors made by GPT-5

The diagram generated by GPT-5 is not completely correct. For instanc,e the text "stratified splits" should be placed along with "k-fold CV". Tokenization was not the precursor in "classical" ML mdels. The term "classical" was never mentioned in the manuscript but was adopted when the manuscript mentions "traditional ML" throughout.

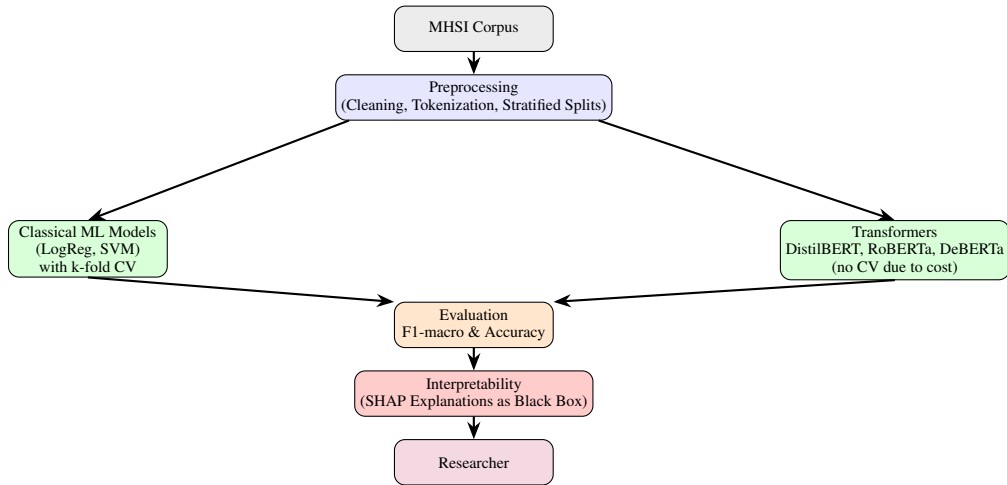

## F   AI Research Autonomy / AI Contribution Disclosure

AI systems' role: An AI system (transformer language model + training script) performed the bulk of model development, hyperparameter sweeps, metric computation, and figure generation. Humans provided dataset curation, labeling guidelines, final labeling oversight, experimental design decisions, and final manuscript editing. The AI is not listed as a human author; humans are the corresponding authors but we document AI contributions in the checklist as required. Agents for Science

### F.1   Responsible AI Statement (concise)

We followed NeurIPS/standard ethical guidelines and considered risks from automated stigma detection. Key actions: (1) human oversight for labeling and final decisions, (2) transparency via per-instance SHAP explanations to support human review, (3) dataset de-identification and adherence to platform terms of service, and (4) a discussion of potential harms (false positives leading to censorship; false negatives perpetuating harm) and mitigation strategies, including human-in-the-loop workflows and threshold tuning to prioritize recall/precision depending on downstream use.

### F.2   Reproducibility Statement

We provide code for training, evaluation, checkpoints, and the SHAP explanation notebook. Exact package versions, random seeds, and hardware (GPU type) are listed on the accompanying GitHub. We include a script to reproduce reported metrics given the provided checkpoint and test set.


